# The Liver Surface Is an Attractive Transplant Site for Pancreatic Islet Transplantation

**DOI:** 10.3390/jcm10040724

**Published:** 2021-02-12

**Authors:** Akiko Inagaki, Takehiro Imura, Yasuhiro Nakamura, Kazuo Ohashi, Masafumi Goto

**Affiliations:** 1Division of Transplantation and Regenerative Medicine, Graduate School of Medicine, Tohoku University, Sendai 980-0872, Japan; akiko.inagaki.b1@tohoku.ac.jp (A.I.); takehiro.imura.b1@tohoku.ac.jp (T.I.); 2Division of Pathology, Faculty of Medicine, Tohoku Medical and Pharmaceutical University, Sendai 983-8536, Japan; yasu-naka@tohoku-mpu.ac.jp; 3Graduate School of Pharmaceutical Sciences, Osaka University, Osaka 565-0871, Japan; ohashikazuo33@gmail.com

**Keywords:** islet transplantation, transplantation site, type 1 diabetes mellitus, cellular therapy, engraftment

## Abstract

In the current clinical islet transplantation, intraportal transplantation is regarded as the gold-standard procedure. However, in this procedure, 50 to 70% of the transplanted islets are immediately damaged due to a strong innate immune response based on islet–blood contact. We investigated the transplant efficiency of a novel method of liver surface transplantation using a syngeneic keratinocyte sheet to avoid islet–blood contact. To examine the influence of the keratinocyte sheet, substantial amounts of syngeneic islets (8 IEQs/g) were transplanted on the liver surface of diabetic rats, while marginal amounts of islets (4 IEQs/g) were transplanted via intraportal transplantation to compare the transplant efficiency. Blood glucose, intraperitoneal glucose tolerance, immunohistochemistry, and in vivo imaging findings of the cell sheet were evaluated. The study showed that islet transplantation to the liver surface immediately followed by a syngeneic keratinocyte sheet covering was effective for curing diabetic rats, while no rats were cured in the group without the cell sheet. Notably, islet grafts transplanted via this approach appeared to penetrate into the liver parenchyma. However, the transplant efficiency did not reach that of intraportal transplantation. Further refinements of this approach by introducing mesothelial or fibroblast cell sheets in combination with a preferable scaffold for islet grafts may help to improve the transplant efficiency.

## 1. Introduction

Pancreatic islet transplantation has become an established treatment for severe type 1 diabetic patients [1,2,3]. At present, intraportal islet injection is regarded as an established procedure for clinical islet transplantation. However, this method has several drawbacks, including the risk of portal vein embolism [4] and the occurrence of strong innate immune reactions [5]. After intraportal islet transplantation 50 to 70% of islet grafts are immediately destroyed due to the instant blood-mediated inflammatory reaction (IBMIR) characterized by the activation of both coagulation and complement cascades [6,7,8,9]. Furthermore, the nuclear protein high-mobility group box 1 (HMGB-1) released from the destroyed islets is reported to enhance the damages on islets per se through the innate immune systems [10].

Interrupting islet–blood contact to avoid IBMIR may be one way to improve engraftment efficiency in intraportal islet transplantation. Notably, the islet cell-endothelial cell complex produced by culturing human islet cells and arterial endothelial cells does not induce IBMIR when in contact with blood, mostly due to the effective interruption of islet–blood contact by the endothelial cell component [11,12]. The development of approaches involving islet transplantation to a new site wherein islet–blood contact can be avoided is therefore eagerly anticipated.

The renal subcapsular space definitely meets the above requirement for a site free from islet–blood contact. In the field of pancreatic islet transplantation, this site has long been regarded as one of the most favorable transplant sites, presumably because islet–blood contact can be avoided [13,14,15]. However, invasive renal subcapsular transplantation is not suitable for islet transplant candidates; as such, severely diabetic patients usually have diabetic nephropathy. Furthermore, islet transplantation into the renal subcapsular space is not physiological, since insulin secreted from the islets is metabolized in the liver. Therefore, this transplant site is well-recognized as an experimental procedure by most researchers in the field [7].

However, the liver surface can be accessed via a laparoscopic approach, making it a minimally invasive procedure. This site is unrelated to nephropathy and is expected to effectively prevent islet–blood contact if the procedure is well adjusted. Unlike the renal subcapsular space, this site is physiological in terms of insulin metabolism. Furthermore, the space is much larger than the renal subcapsular space, which seems preferable in the clinical setting, as huge amounts of tissues are often required for implantation in clinical cases. Thus, the liver surface is a potential candidate site for islet transplantation.

Indeed, Fujita et al. reported that a single-cell islet cell sheet transplanted to the liver surface was effective in a mouse model [16]. Given that a considerable number of islet cells are lost in the preparation process of the islet sheet, this approach still needs substantial improvements before clinical application. Likewise, Yamasaki et al. previously reported that the blood glucose of diabetic rats was normalized as a result of islet transplantation to the liver surface using a chitin film, and islet engraftment in the liver parenchyma was demonstrated [17]. However, this procedure caused severe adhesion between the diaphragm and the liver surface, mostly based on the foreign body reactions of the chitin film, resulting in the demand for huge amounts of islets to cure diabetic rats.

Therefore, in the present study, we proposed a novel method of liver surface transplantation in which islets were completely covered with a syngeneic keratinocyte sheet to circumvent foreign body responses. In the present study, we used the keratinocyte sheet for islet transplantation because it was easy to collect the skin as a raw material, the producing method was already established, and the strength of the keratinocyte sheet was robust. We first evaluated the effects of a syngeneic keratinocyte sheet on the islet engraftment in the liver surface transplantation procedure. We then investigated the transplant efficiency between the abovementioned method and the current standard method of intraportal islet transplantation.

## 2. Materials and Methods

### 2.1. Animals

All of the animals used in the present study were handled in accordance with the Guide for the Care and Use of Laboratory Animals published by the National Institutes of Health, and the guidelines for animal experiments and related activities at Tohoku University (approved protocol ID: 2013NiA-008). Male Lewis Rats were used as recipients and islet donors (8 weeks of age; Japan SLC Inc., Shizuoka, Japan). Male and female Lewis rats (1 to 3 days of age) were used as keratinocyte donors. On in vivo imaging, keratinocytes were obtained from male and female luciferase transgenic Lewis rats (1 to 3 days of age) provided by Eiji Kobayashi (Keio University) and bred at Tohoku University. All surgeries were performed under general anesthesia, and all efforts were made to minimize suffering.

### 2.2. Fabrication of the Keratinocyte Sheet

The keratinocyte sheet was prepared by culturing keratinocytes on a collagen gel containing fibroblasts. Fibroblasts were isolated from the skin of 1- to 3-day-old female or male Lewis rats. The tissue was cut into small pieces and then harvested with 0.25% trypsin and 1 mM ethylenediaminetetraacetate (EDTA) solution (Thermo Fisher Scientific, Waltham, MA, USA) for 30 min at 37 °C to digest the tissue. Digestive tissue was filtered through a 70 µm nylon mesh and centrifuged at 350 g for 5 min. The separated cells were seeded in a cell culture flask and cultured (Thermo Fisher Scientific) in Dulbecco’s modified Eagle’s medium (DMEM; Sigma Aldrich, St. Louis, MO, USA) containing 10% fetal bovine serum (FBS; Hycult Biotech, Wayne, PA, USA) and penicillin-streptomycin (Thermo Fisher Scientific) at 37 °C in 5% CO_2_. Fibroblasts from passages 3 to 5 were used to prepare keratinocyte sheets.

Neutralized 0.2% collagen solution (3 mL; Nitta Gelatin Inc., Osaka, Japan) containing 2 × 10^5^ cells/mL of fibroblasts was gelled in a 6-well cell culture insert (Thermo Fisher Scientific) and then in DMEM containing 10% FBS. After being incubated for 1 day, keratinocytes were seeded on the collagen gel. Keratinocytes were isolated from the skin of 1- to 3-day-old Lewis or luciferase transgenic rats. Skin tissue was immersed in a phosphate-buffered saline solution containing 1.2 U/mL Dispase (Thermo Fisher Scientific) at 4 °C for 18 to 21 h. The skin was then exfoliated into the dermis and epidermis in 0.25% Trypsin-EDTA, cut into small pieces with scissors, and incubated for 10 min at 37 °C. Digestive tissue was filtered through a 70 µm nylon mesh and centrifuged at 350 g for 5 min, and pellets were suspended in keratinocyte serum-free medium (Thermo Fisher Scientific). A total of 3 × 10^6^ keratinocytes were seeded on the fibroblast gel and cultured in keratinocyte medium for one day before being further cultured in differentiation medium (keratinocyte medium: DMEM: FBS = 1:1:0.1 ratio) containing 50 µg/mL L-ascorbic acid (Sigma-Aldrich, St. Louis, MO, USA) for 4 days. The keratinocyte sheets stripped from the collagen gel were used for liver surface transplantation (Figure 1A).

### 2.3. The Evaluation of the Engraftment of Keratinocyte Sheets Transplanted onto the Liver Surface

The IVIS Spectrum (PerkinElmer, Waltham, MA, USA) was used to detect the luciferase expression. The keratinocyte sheet from luciferase transgenic rats with the support membrane was transplanted onto the right median lobe of the liver surface (*n* = 3), and the support membrane was removed. For in vivo imaging, 150 mg/kg D-luciferin potassium salt (Promega Corp., Madison, WI, USA) was intraperitoneally injected. One minute after injection, bioluminescence images were captured for 3 min. Regions of interest (ROIs) of the same area of the captured image were analyzed, and total flux (pixel/s) was quantified using the Living Image software (PerkinElmer). To confirm the presence of transplanted keratinocyte sheet, imaging was performed at 1, 3, 7, 14, 21, and 28 days after transplantation. In addition, recipient livers were harvested 28 days after transplantation, and then hematoxylin-eosin (HE) staining and cytokeratin AE1/AE3 staining were performed.

### 2.4. The Induction and Diagnosis of Diabetes in the Recipients

Diabetes was induced by the intravenous injection of 70 mg/kg streptozotocin (Sigma-Aldrich). Rats whose non-fasting blood glucose levels were ≥400 mg/dL on the transplant day were considered diabetic.

### 2.5. Islet Isolation and Transplantation

Rat islet isolation and culturing were performed as previously described [18]. The islets were cultured in Roswell Park Memorial Institute-1640 medium (Thermo Fisher Scientific) containing 5.5 mmol/L glucose and 10% FBS at 37 °C in 5% CO_2_ and humidified air overnight before transplantation.

In the liver surface transplant procedure, 8 islet equivalents (IEQs)/g body weight were placed onto the surface of the right median lobe and covered with a keratinocyte sheet (Cell sheet group, *n* = 4) or left without one (Control group, *n* = 4) (Figure 1B).

### 2.6. Intravenous Glucose Tolerance Test

An intravenous glucose tolerance test (IVGTT) was performed 28 to 35 days after islet transplantation as previously described [19]. D-glucose (3.0 g/kg) was intravenously infused, and the blood glucose concentrations were determined before and at 5, 10, 20, 30, 60, 90, and 120 min after the injection of glucose.

### 2.7. Immunohistochemical Staining

The keratinocyte sheet and recipient liver at 38 days after transplantation were fixed with 4% paraformaldehyde and embedded in paraffin for immunohistochemical staining. Immunohistochemical staining was performed using anti-cytokeratin AE1/AE3 antibody (MerckMillipore, Darmstadt, Germany), anti-insulin antibody (Dako, Glostrup, Denmark), and an Envision kit (Dako). Elastica-Masson staining was performed to selectively dye collagen fibers on the liver surface.

### 2.8. The Comparison of Islet Engraftment between the Liver Surface Transplantation with Keratinocyte Sheets and Intraportal Transplantation

For intraportal transplantation (Intraportal group, *n* = 4), 4 IEQs/g body weight of islets in a total volume of 500 µL were infused into the recipient liver through the portal vein using a 25-gauge surshield (TERUMO BCT, Inc., Tokyo, Japan).

### 2.9. Statistical Analyses

All data are expressed as the mean ± standard error of the mean (SEM). Statistical significance was determined using Student’s *t*-test or two-way ANOVA. *p* values of < 0.05 were considered to indicate statistical significance.

## 3. Results

### 3.1. Engraftment of the Keratinocyte Sheet Transplanted onto the Liver Surface

The keratinocyte sheets transplanted onto the liver surface were engrafted for at least 28 days after transplantation. It was found that the area of the keratinocyte sheets transplanted on the liver surface decreased on 28 days after transplantation. The total flux of the ROI of the liver was 6.60 × 10^6^ ± 0.82 × 10^6^ (p/s) and 1.16 × 10^6^ ± 0.51 × 10^6^ (p/s) at 1 and 28 days after transplantation (Figure 2A).

### 3.2. Influence of the Keratinocyte Sheet on Islet Engraftment in Liver Surface Transplantation

Liver surface transplantation without a keratinocyte sheet (control group) maintained high blood glucose levels throughout the entire study period. In contrast, liver surface transplantation with a keratinocyte sheet (cell sheet group) resulted in normalized blood glucose levels within a few days after transplantation (Figure 3A). Furthermore, the body weight in the cell sheet group was significantly higher than in the control group (Figure 3B).

### 3.3. IVGTT Results

The glucose tolerance was significantly ameliorated in the cell sheet group compared with the control group (area under the curve [AUC]: 22,889 ± 3839 vs. 48,404 ± 4635 min·mg/dL, *p* < 0.01, Kg values: 1.59 ± 0.48 vs. 0.55 ± 0.27%/min, *p* < 0.01) (Figure 4A–C).

### 3.4. Immunohistochemical Staining of the Transplanted Islets and Keratinocyte Sheets on the Liver Surface

The transplanted keratinocyte sheets and islets were evaluated by cytokeratin AE1/AE3 and AE3 staining (Figure 5A,B). Some of the transplanted islets were detected in the liver parenchyma. The transplanted keratinocyte sheets on the liver surface formed an atheroma-like mass in the liver at a location distant from the islets. Furthermore, the engrafted islets were found to be present either in the hepatic capsule or in the liver parenchyma (Figure 6A–C).

### 3.5. The Comparison of Islet Engraftment between Liver Surface Transplantation with Keratinocyte Sheets and that via Intraportal Transplantation

In the case of marginal amounts of islet transplantation, the blood glucose levels were significantly higher in the liver surface group than in the intraportal group (Figure 7A). In addition, the body weight of the recipients was also higher in the intraportal group than in the liver surface group (Figure 7B).

## 4. Discussion

The present study showed that islet transplantation to the liver surface immediately followed by a syngeneic keratinocyte sheet covering was effective for curing diabetic rats. In this procedure, islet grafts were transplanted onto the liver surface without making contact with the blood and immediately engrafted, resulting in normalization of the blood glucose levels throughout the entire observation period. However, while this approach required smaller amounts of islets to cure diabetic animals than liver surface transplantation with a chitin film, the transplant efficiency did not reach that of the current gold standard, intraportal transplantation, when the same amounts of islets were implanted.

Of particular interest, the islet grafts transplanted via the abovementioned approach appeared to penetrate the liver parenchyma (Figure 5A and Figure 6A–C). It is well known that islet grafts transplanted into the kidney subcapsular space tend to not stay within the subcapsular space but penetrate into the kidney parenchyma, which is preferable for islet engraftment since the vascular network is abundant in the parenchyma [20,21]. In contrast to renal subcapsular transplantation, however, the islet grafts must pass through the liver capsule to place in liver parenchyma, since islets are implanted on the liver capsule rather than under the capsule in the proposed approach. The detailed mechanism underlying why the islets pass through the liver capsule is unclear at present. One possible explanation is that the liver capsule becomes coarse due to continuous mechanical stimulation caused by islets and/or keratinocyte sheets. This concept is partially consistent with the finding of a previous report [17]. Another possible explanation is that some endogenous enzymes, including matrix metalloproteinase-9, derived from the islet grafts may contribute to the islets passing through the liver capsule [22]. We recently showed that the hepatocyte function seemed to improve by co-culturing and/or co-transplantation with pancreatic islets [23]. Considering this novel finding, it is speculated that the hepatocytes located near the transplanted islet grafts may have received benefits from them, subsequently resulting in the establishment of a preferable milieu for islet engraftment as well. Given these findings, islet transplantation to the liver surface can be considered a feasible procedure. However, further investigations are needed to clarify the precise mechanisms underlying this phenomenon, a point we intend to explore in our next study.

In this study, isolated islets were transplanted onto the liver surface using a syngeneic keratinocytes to mitigate the foreign body reactions that were observed in the case of chitin film application [17]. As expected, this novel approach made it possible to normalize the blood glucose levels of diabetic rats using smaller numbers of islet grafts than liver surface transplantation with a chitin film (Figure 3) (8 IEQs/g vs. 12–14 IEQs as a conversion value/g) [17]. However, unexpectedly, strong adhesion between the diaphragm and the liver was observed after islet transplantation, even in the proposed approach. Furthermore, the keratinocyte sheet was not present on the liver surface, instead infiltrating into the liver parenchyma and forming an atheroma-like mass at approximately one month after islet transplantation (Figure 6B). Notably, when the keratinocyte sheet alone was transplanted onto the liver surface without islet grafting, the keratinocytes formed multiple layers on the liver surface and neither penetrated into the liver parenchyma nor adhered to the diaphragm (Figure 2B,C). As islet grafts have the property of penetrating into the kidney parenchyma following renal subcapsular transplantation and a similar tendency was observed with liver surface transplantation, the abovementioned unexpected finding may be attributed to the keratinocyte sheet attached to the islets also penetrating the liver parenchyma along with the islet grafts. As the liver capsule consists of a serosal membrane and fibrous connective tissue and no keratinocytes are present on the liver surface, differences in cell components between the cell sheet and liver surface may have induced inflammatory reactions, consequently resulting in the occurrence of adhesion and formation of an epidermoid cyst. Therefore, a serosal mesothelial cell or a fibroblast (constituent cells of the liver capsule) may be a suitable candidate cell source for constructing cell sheets to cover the islet grafts. The use of these cell sources may efficiently avoid adhesion between the diaphragm and liver surface, subsequently making it possible to repeat islet implantation onto the liver surface if necessary. Likewise, Cui et al. recently reported that islet transplantation on the liver surface, in which a combination of human amniotic membrane (HAM) and human amniotic epithelial cells was applied to islet transplantation to the liver surface, was effective in a xenogeneic mouse model [24]. Although the transplant efficiency between this approach and intraportal transplantation was not examined in this study, considering the immunologically potential advantages of this procedure, the HAM can be a candidate cell source for producing cell sheets as well.

In the present study, small-sized islets appeared to infiltrate the liver parenchyma, while large-sized islets remained under the thickened hepatic capsule space (Figure 6A–C). This interesting finding is consistent with a previous report in which small islets were more effective for normalizing blood glucose levels than large islets in renal subcapsular transplantation to diabetic rats [25,26]. These results suggest that the size of islets may influence the success rate of islet transplantation, due not only to the induction of central necrosis in large islets but also enhanced penetration into the parenchyma in small islets.

## 5. Conclusions

The present study showed that islet transplantation to the liver surface immediately followed by a syngeneic keratinocyte sheet covering is effective for curing diabetic rats. Further refinements to this approach, such as by introducing mesothelial or fibroblast cell sheets in combination with a preferable scaffold for islet grafts, may be a promising strategy for improving transplant efficiency.

## Figures and Tables

**Figure 1 jcm-10-00724-f001:**
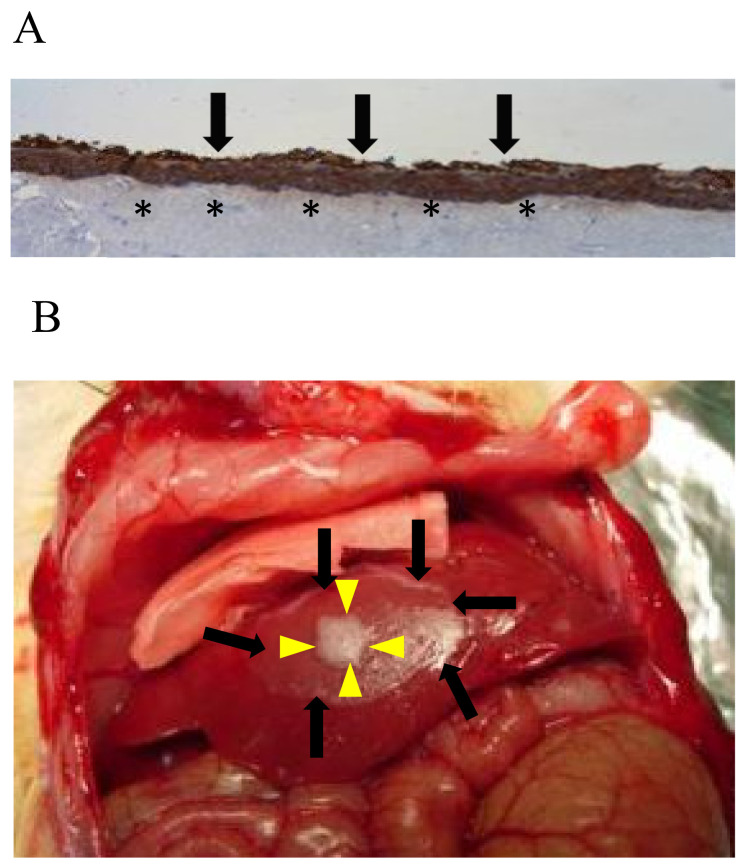
The cytokeratin AE1/AE3 staining of the keratinocyte sheet used in the present study (**A**), and photograph of transplanted rat islets on liver surface using the keratinocyte sheet (**B**). Keratinocyte sheets (black arrows) cultured on collagen gel containing fibroblasts (asterisks) were placed on the liver surface to cover islet grafts (yellow arrows).

**Figure 2 jcm-10-00724-f002:**
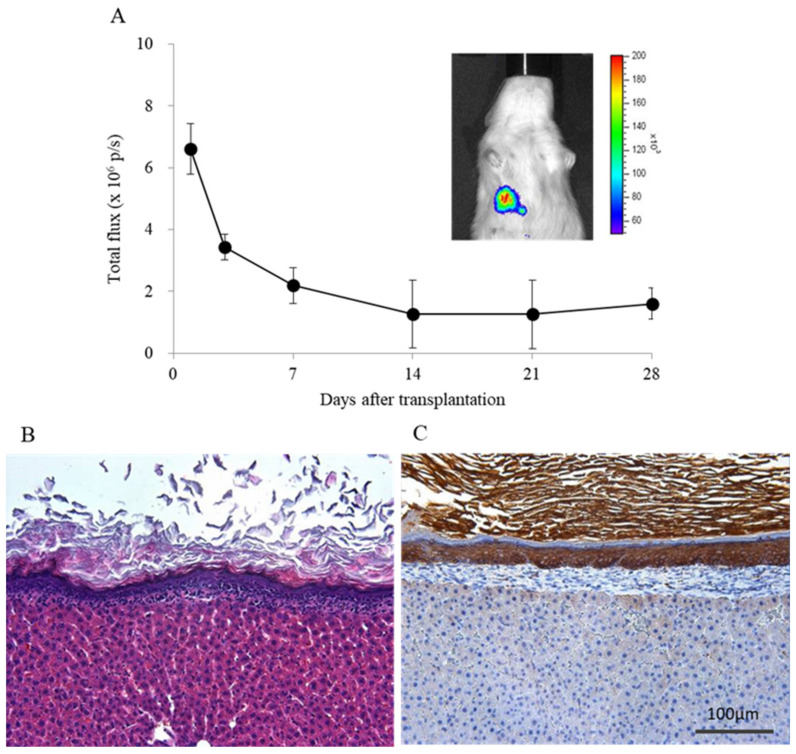
Engraftment of keratinocyte sheets transplanted on liver surface. (**A**) Bioluminescence imaging of keratinocyte sheet implanted on liver surface. The keratinocytes transplanted on liver surface were proved to survive until 28 days after transplantation (*n* = 3). (**B**) The hematoxylin and eosin staining and (**C**) cytokeratin AE1/AE3 staining of keratinocyte sheet transplanted on liver surface at 28 days after transplantation.

**Figure 3 jcm-10-00724-f003:**
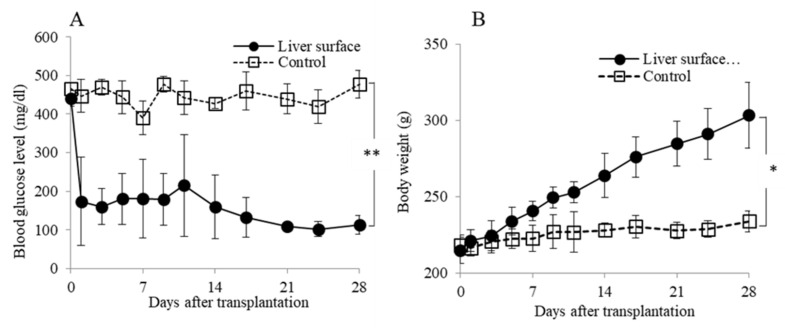
Metabolic evolution after islet transplantation. (**A**) The changes in the blood glucose levels after islet transplantation and (**B**) the changes in body weight after islet transplantation. The liver surface transplantation with keratinocyte sheet (cell sheet group; black circle, *n* = 4) showed significantly better blood glucose levels than without keratinocyte sheet (control group; white square, *n* = 4). Data are mean ± SEM (** *p* < 0.01, * *p* < 0.05, respectively).

**Figure 4 jcm-10-00724-f004:**
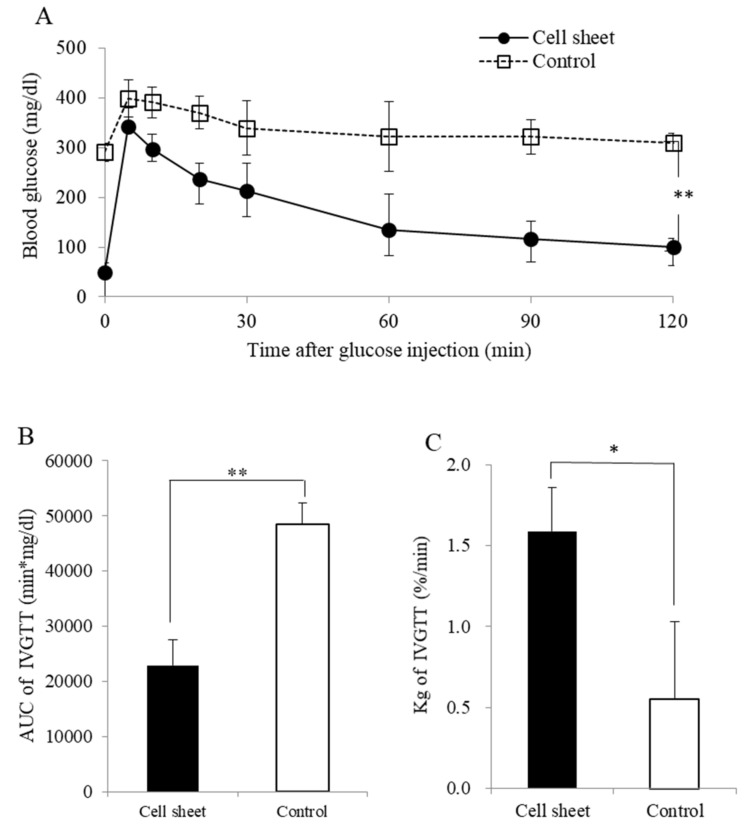
Profile of glucose tolerance of the cell sheet and control groups. (**A**) The blood glucose changes of the intravenous glucose tolerance test (IVGTT) (cell sheet group; black circle, *n* = 4, control group; white square, *n* = 4). (**B**) The area under the curve (** *p* < 0.01) and (**C**) Kg values (* *p* < 0.05) of the IVGTT.

**Figure 5 jcm-10-00724-f005:**
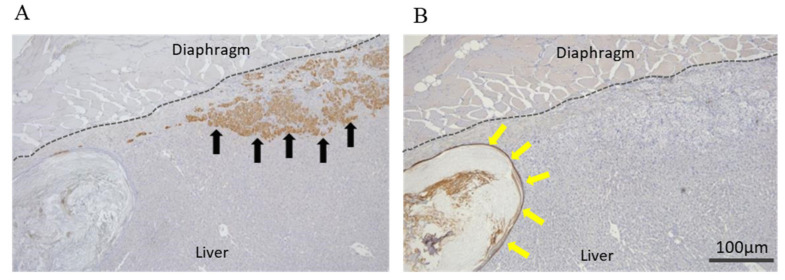
Histological analyses of transplanted islets and keratinocyte sheet on liver surface in the cell sheet group. (**A**) Insulin and (**B**) Cytokeratin AE1/AE3 immunohistochemical staining. The transplanted islets penetrated into the liver parenchyma (black arrows). The keratinocyte sheets that covered liver surface formed an atheroma-like mass in the liver (yellow arrows). The dotted lines indicate the boundary between the liver and the diaphragm.

**Figure 6 jcm-10-00724-f006:**
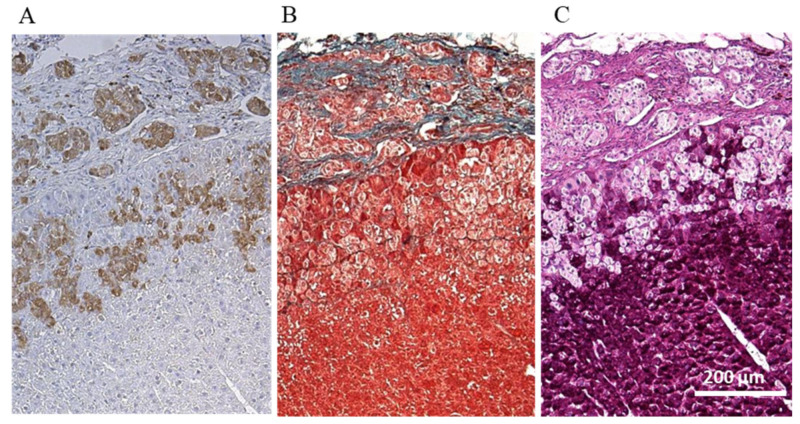
Histological analyses of islets engrafted on liver surface. (**A**) Insulin (brown), (**B**) Elastica-Masson staining to clearly distinguish connective tissues such as elastic fibers (grayish green), and (**C**) Periodic acid-Schiff staining to identify the liver parenchyma (purple).

**Figure 7 jcm-10-00724-f007:**
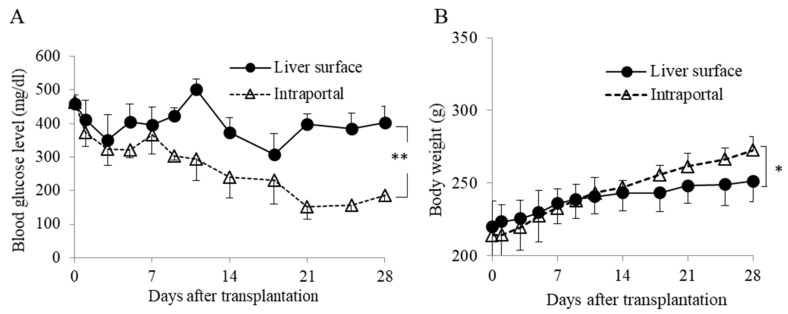
The outcome of islet (4 IEQs/g) engraftment after transplantation in the liver surface and intraportal groups. The changes in the blood glucose levels after islet transplantation (**A**) and the changes in body weight (**B**) after islet transplantation. The intraportal transplantation (intraportal group; white triangle, *n* = 4) showed significantly better blood glucose levels than liver surface transplantation with keratinocyte sheet (liver surface group; black circle, *n* = 4). Data are mean ± SEM (** *p* < 0.01, * *p* < 0.05, respectively).

## Data Availability

The data that support the findings of this study are available from the corresponding author, M.G., upon reasonable request.

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
