# Peer review of "The Liver Surface Is an Attractive Transplant Site for Pancreatic Islet Transplantation"

_jcm, 2021, doi:10.3390/jcm10040724_

Round 1

Reviewer 1 Report

The aim of this study is to find alternative sites and methods of pancreatic islet transplantation, which could improve efficacy of clinical pancreatic islet transplantation. Authors used the keratinocyte sheet cells collected from syngeneic rats and used this to cover transplanted islets on the liver surface. This method improved islet graft survival and correct the metabolic function of STZ induced diabetic rats. However, the outcome of transplantation when used marginal dose of islets was still better in the gold standard method, i.e. portal vein injection than in the experimental method using keratinocyte sheet.

This is the first article to demonstrate the keratinocyte sheet covered islets on the liver surface could improve metabolic control of diabetic recipients. Authors also showed several interesting findings, such as the keratinocyte sheet itself did not cause inflammation on liver surface, however, transplanted pancreatic islets under the keratinocyte sheet induce inflammation and migration to the liver parenchyma. Though the numbers of experiments were few, the conclusion was clear.

There are a few facts, the authors should consider in revised version:

  1. In introduction:

It will be helpful with  an introduction on why authors decided to use keratinocyte sheet for pancreatic islet transplantation and what is known about function, immunological activities, tumorigenicity and etc.

  1. Methods:

Explain why authors used 1-3 days old rats as source of fibroblast and keratinocytes instead of the same age of transplanted rat (8 weeks). For an autologous setting, it is more reasonable to use age- matched rats as source of cells.

Autologous means itself. If the tissues are collected from the rats with genetically matched individuals, this should be called as syngeneic.

  1. Results:

Figure 2A: Total flux reduced quickly during the first 14 days and then stabilized. In the discussion, authors mentioned that keratinocytes growth forms multiple layers without penetration to liver surface or adhesion to diaphragm. Was the area of sheet diminished during 28 days? Did authors find any sign of keratinocyte expansion?

Authors should explain about Figure 2 B and C as text in the results.

  1. Discussion:

4.1, Authors commented: “The detailed mechanism underlying why the islets pass through the liver capsule is unclear at present.” Beta cells need high oxygen supply and therefore it is important to get vascularization inside the grafts quickly. The grafts are exposed to hypoxia and malnutrition immediately after transplantation. In this manuscript, authors did not mention about the endothelial cell growth inside the grafts nor the keratinocyte sheet. It would be interesting to show how the blood supply forms. To add endothelial cell staining and discuss how the capillary network is established with or without islet grafts may give some information.

4.2, In the following sentence authors commented as ‘ One possible explanation is that the liver capsule becomes coarse due to continuous mechanical stimulation caused by islets and/or keratinocyte sheets. ’ The transplantation site beside diaphragm may have more movement due to the animal breathing. Do authors consider transplanting islets at the non-diaphragm site of liver?

4.3, In figure 6, authors showed that big islets stay on the top of liver surface and smaller islets were detected in the liver parenchyma. Are they migrated or newly divided? It would be informative if authors add proliferation marker staining such as Ki67 to evaluate whether there is any cell division in the migrated islet cells. The possibility of cell division or differentiation could be discussed.

Minor points: A few typos and a few suggestions

Introduction:

  1. Page 1 Introduction 1st paragraph Line 5: After intraportal islet transplantation, the transplanted islet grafts are … this part can be expressed as a large proportion of islet grafts or 50-70 % of islet grafts are…

Methods

  1. Page 2, Methods 2.1 line 9: ‘All surgeries were performed general anesthesia’ could be All surgeries were performed under general anesthesia
  2. Page 3 line 4: Dulbecco’s modified Eaglel’s medium correct to Eagle’s

Results

  1. Page 4, ROI: first appearance of acronym should be spelled out
  2. Figure 6 B figure legend: B) Elastica Masson staining to clearly distinguish connective tissues such as elastic fibers (green) : It does not look green. Grey?

Discussion

  1. page 10 line 3-5 ‘As expected, this novel approach made it possible to normalize the blood glucose levels of diabetic rats using smaller numbers of islet grafts than liver surface transplantation with a chitin film (Figure 3) [17] (8 IEQ/g vs ….).’ If applicable, add information of pancreatic islet number that were used in both studies .
  2. page 10 line 5 remove ‘rather’
  3. page 10 line 9 athenoma should be atheroma
  4. page 10 line 12 Figure 1 should be Figure 2
  5. Page 10: ‘5. Concludions’ should be 5. Conclusions

Reviewer 2 Report

In this paper authors intend to study the liver surface as transplantation site. Reviewer has following comments and questions:

  • Kindly explain and discuss the rational for islet dose selection in each arm i.e. surface with or without K sheath and intraportal (IP). While it appears that based on previous models of IP implantation, doses are relatively ok or on higher side; but surface implantation doses may be lower for good outcome.
  • It would have been interesting and physiological to compare Liver surface group with an invasive renal subcapsular group. Please discuss.
  • Why such low number of animals selected for study? Define rational or limitation (e.g. Financial constrains etc)
  • Methods number 2.7: Make sure you meant 38 days and not 28 based on description above
  • Previously very successful liver surface implantation with help of amniotic epithelial membrane as sheet has been described. Should discuss pros, cons, comparison with that approach citing the article in discussion [Transplant 2020 Apr;52(3):982-986]
  • Discussion should have a paragraph of limitation of study, future direction, and alternative approach
